# Hepatoprotective Effects of Four Brazilian Savanna Species on Acetaminophen-Induced Hepatotoxicity in HepG2 Cells

**DOI:** 10.3390/plants12193393

**Published:** 2023-09-26

**Authors:** Gislane dos Santos Ribeiro, Diegue Henrique Nascimento Martins, João Victor Dutra Gomes, Noel William Davies, Christopher William Fagg, Luiz Alberto Simeoni, Mauricio Homem-de-Mello, Pérola Oliveira Magalhães, Dâmaris Silveira, Yris Maria Fonseca-Bazzo

**Affiliations:** 1Pharmacy Department, Health Sciences School, University of Brasília, Brasilia 70910-900, Brazil; gss.ribeiro@hotmail.com (G.d.S.R.); diegue.hen@gmail.com (D.H.N.M.); joao.gomes@unb.br (J.V.D.G.); lsimeoni@unb.br (L.A.S.); mauriciohmello@unb.br (M.H.-d.-M.); perolam@hotmail.com (P.O.M.); damaris@unb.br (D.S.); 2Central Science Laboratory, University of Tasmania, Hobart, TAS 7005, Australia; noel.davies@utas.edu.au; 3Department of Botany, Institute of Biological Science, University of Brasília, Brasilia 70910-900, Brazil; fagg@unb.br

**Keywords:** acetaminophen, hepatotoxicity activity, hepatoprotective activity

## Abstract

We investigated four Cerrado plant species, i.e., *Cheiloclinium cognatum* (Miers) A.C.Sm, *Guazuma ulmifolia* Lam., *Hancornia speciosa* Gomes, and *Hymenaea stigonocarpa* Mart. ex Hayne, against acetaminophen toxicity using an in vitro assay with HepG2 cells. The activity against acetaminophen toxicity was evaluated using different protocols, i.e., pre-treatment, co-treatment, and post-treatment of the cells with acetaminophen and the plant extracts. HepG2 cell viability after treatment with acetaminophen was 39.61 ± 5.59% of viable cells. In the pre-treatment protocol, the extracts could perform protection with viability ranging from 50.02 ± 15.24% to 78.75 ± 5.61%, approaching the positive control silymarin with 75.83 ± 5.52%. In the post-treatment protocol, all extracts and silymarin failed to reverse the acetaminophen damage. In the co-treatment protocol, the extracts showed protection ranging from 50.92 ± 11.14% to 68.50 ± 9.75%, and silymarin showed 77.87 ± 4.26%, demonstrating that the aqueous extracts of the species also do not increase the toxic effect of acetaminophen. This protection observed in cell viability was accompanied by a decrease in ROS. The extracts’ hepatoprotection can be related to antioxidant compounds, such as rutin and mangiferin, identified using HPLC-DAD and UPLC-MS/MS. The extracts were shown to protect HepG2 cells against future APAP toxicity and may be candidates for supplements that could be used to prevent liver damage. In the concomitant treatment using the extracts with APAP, it was demonstrated that the extracts do not present a synergistic toxicity effect, with no occurrence of potentiation of toxicity. The extracts showed considerable cytoprotective effects and important antioxidant characteristics.

## 1. Introduction

The liver, an organ responsible for metabolism, detoxification, and excretion, is predisposed to xenobiotics and, therefore, is a tissue susceptible to toxicity, which can cause morphological and functional changes [1,2]. The consumption of medicinal plants and products based on medicinal plants is high, especially in developing countries, and is increasing significantly due to the influence of empirical knowledge passed down from generation to generation and public interest in natural therapies [3]. This consumption is made with the belief of safety because they are natural products [3].

Plants contain multiple compounds and present a risk of causing liver damage. However, there is insufficient information on the compounds’ toxicological and pharmacological profiles. In addition, several cases of intoxication are underreported [4]. It is, therefore, important to study the composition of medicinal plants to identify compounds with hepatotoxic potential.

On the other hand, medicinal plants can also have beneficial compounds for discovering new medicines [3]. Studies have shown that secondary metabolites such as polyphenols, anthraquinones, terpenes, and sulforaphane can activate the hepatocyte antioxidant defense system with Nrf2 as the core player, reduce oxidative stress damage, and protect the liver [5].

Non-opioid analgesics and non-steroidal anti-inflammatory drugs (NSAIDs) correspond to the most commonly used non-prescription drugs, the so-called over-the-counter medicines (OTC) [6]. These OTCs make access easy, which also threatens patient safety by favoring the risk of unintentional intoxication [7]. Acetaminophen or N-acetyl-p-aminophenol (APAP) is a drug of the non-opioid analgesic class widely used due to its antipyretic, analgesic, and anti-inflammatory action [8]. It is found in several pharmaceutical forms, as a component of several over-the-counter drugs, and is widely used due to its ease of access and low cost. It is considered safe in therapeutic doses. However, improper use presents a hepatotoxicity risk related to overdose [9]. Many cases of acetaminophen overdose in the USA are related to acute liver failure [7]. Moreover, the second most common cause of liver transplantation worldwide is acetaminophen [10].

Since then, the safety and efficacy of plant extracts and plant-derived compounds have been investigated as monotherapy or adjuvant to conventional drugs to limit the toxicity induced by acetaminophen [11].

In this context, considering that Brazil is a major holder of biodiversity and the Cerrado biome, with irreplaceable endemic species, and is considered one of the 25 hotspots of this biodiversity, it is crucial to identify whether plant species are hepatotoxic or biologically promising for the development of pharmaceutical alternatives. Thus, four Cerrado plant species from the Federal District region (Brazil) were selected for this study: *Cheiloclinium cognatum* (Miers) A.C.Sm., *Guazuma ulmifolia* Lam., *Hancornia speciosa* Gomes, and *Hymenaea stigonocarpa* Mart. Ex Hayne.

The species *C. cognatum* belonging to the Celestraceae family is known as “Bacupari” or “Pitombinha”. Its leaves are popularly used to treat fever and edema [12,13]. Chemical evaluation of the roots demonstrated the presence of triterpenes, some of which showed free radical-scavenging activity [14].

*Guazuma ulmifolia* belongs to the Malvaceae family and is popularly known as “Araticum-Bravo”, “Mutamba”, or “Chico-Magro” [15]. The most reported biological activities for this species are antimicrobial, antioxidant, and antiprotozoal, and there is a scarcity of toxicity studies of *G. ulmifolia* extracts and their isolated compounds [16].

*Hancornia speciosa* belongs to the Apocynaceae family, popularly known as “Mangaba”. The leaves are traditionally used to treat wounds and inflammatory processes, with this use supported by evidence [17]. Other activities for *H. speciosa* leaves have been reported, such as antioxidant, antihypertensive, cytotoxic, and acetylcholinesterase inhibitory activities [18].

The species *H. stigonocarpa* belongs to the Fabaceae family and is popularly known as “Jatobá do Cerrado”, “Jatobeiro”, or “Jatobá”. Chemical studies have identified phenolic compounds, flavonoids, alkaloids, and coumarins in different plant parts. These have been associated with antioxidant, anti-inflammatory, gastroprotective, antibacterial, and cicatrizing activities [19,20,21,22].

Silymarin was used as a control in this study. It is an extract from milk thistle seeds and has been used to treat hepatic conditions for centuries. Silymarin acts as a free radical scavenger and modulates enzymes associated with developing hepatic damage [23].

Considering the potential of these popularly used species and the lack of hepatotoxicity and hepatoprotection studies, this study aims to evaluate these species’ activity against APAP-induced intoxication.

## 2. Results

### 2.1. Phytochemical Analyses

The obtained extracts presented a final yield ranging from 10.88% to 15.23%. HPLC/DAD and UHPLC/MS/MS techniques were used to identify the chemical composition of the prepared aqueous extracts.

*Cheiloclinium cognatum* (Miers) A.C. Sm. aqueous extract (CCAE) was analyzed with UHPLC/MS/MS. This assay resulted in the identification of a peak of high intensity at the retention time of 2.70 min, which was identified as the xanthanoid mangiferin and another low-intensity peak suggestive of catechin or epicatechin (Figure 1).

The chemical composition of *Hancornia speciosa* Gomes aqueous extract (HEAE) was characterized using HPLC/DAD. Results revealed the presence of peak A with a retention time of 14.50 min showing similarity with chlorogenic acid (similarity index: 0.9998), and peak B with a retention time of 27.57 min showing similarity with hyperoside (similarity index: 0.9964) and isoquercitrin (similarity index: 0.9955) compounds. As hyperoside and isoquercitrin elute at retention times close to that of rutin, a new analysis was performed with the mixture of the extract and the suspected compounds, and the presence of rutin was confirmed (similarity index: 0.9999) (Figure 2).

*Guazuma ulmifolia* Lam aqueous extract (GUAE) was evaluated with HPLC/DAD and UPLC/MS/MS. In the HPLC/DAD analyses, one of the peaks (tR 27.547 min) showed similarity with rutin (similarity index: 0.9973; purity: 1.00; λmax: 256,354). UPLC/MS/MS found ten peaks with flavonoid characteristics. Three peaks, at 4.10, 4.36, and 4.49 min, showed similarity with rutin, presenting M + H and MS/MS fragmentation profile similar to that found in the literature, suggesting the presence of this flavonoid in this aqueous extract (Figure 3 and Figure 4).

*Hymenaea stigonocarpa* Mart. ex Hayne aqueous extract (HSAE) was also evaluated using HPLC/DAD and UHPLC/MS/MS techniques. In the HPLC/DAD analysis, the main peak had flavonoid characteristics, but it was not possible to identify the flavonoid clearly. Further elucidation with UHPLC/MS/MS (Figure 5) showed two peaks presenting an M + H ion and MS/MS fragmentation profile similar to that found in the literature for rutin, suggesting its presence in HSAE. As for the other peaks found, it was not possible to elucidate the structures from the fragmentation profiles (Figure 6).

### 2.2. Cytotoxicity Test

All extracts showed dose-dependent toxicity. GUAE showed the lowest toxicity, and HSAE showed the highest toxicity (Appendix A). CCAE presented IC_50_ of 363.43 ± 69.36 µg/mL. GUAE revealed IC_50_ of 528.97 ± 81.24 µg/mL. HEAE and HSAE showed IC_50_ of 249.97 ± 38.78 µg/mL and 148.37 ± 16.48 µg/mL, respectively. When a viability reduction was greater than 30%, the extract concentrations were considered cytotoxic, as recommended by ISO 10993-5 [24]. Therefore, only concentrations with more than 70% viability were used for APAP-induced toxicity assay.

APAP presented IC_50_ of 13.76 ± 0.84 mmol/L in HepG2 cell viability (Appendix A). Thus, APAP at 15 mmol/L concentration was used to evaluate the activity of extracts against APAP toxicity.

DMSO was used in solution at a concentration of 0.75% as a diluent of APAP. DMSO is widely used for water-insoluble compounds in biological studies. Although it increases their solubility, it presents toxic effects by interacting with metabolism and cell membranes. Thus, the possible interference of DMSO in this assay was evaluated. DMSO did not cause cell death or interference in the assay (Appendix A). There was no cell death observed in concentrations from 0.05% to 1.25%. In contrast, at 2.5% concentration, there was a decrease in cell viability (75.68 ± 4.76%), but not statistically significant compared to control (100% cell viability).

Silymarin showed an IC_50_ of 266.6 ± 28.90 µg/mL in HepG2 cell viability (Appendix A).

### 2.3. Activity against APAP Toxicity

Three treatment protocols were performed: pre-treatment, post-treatment, and co-treatment.

In the pre-treatment protocol, extracts showed protective activity, improving HepG2 cell viability in relation to the APAP group, ranging from 10% to 40%. CCAE presented the highest activity, presenting protection similar to the silymarin (positive control), while HSAE showed the lowest activity (Figure 7).

In the post-treatment protocol, none of the extracts showed reversal activity of APAP toxicity. No extracts improved HepG2 cell viability concerning the APAP group (Figure 8). Compared to the APAP group, silymarin (50 µg/mL) improved significantly in 10.3% of the HepG2 cell viability.

In the co-treatment protocol, the toxicity of APAP did not increase with the presence of the extracts. Extracts improved HepG2 cell viability in relation to the APAP group in a range of 11% to 29%. CCAE showed the best result, improving cell viability by about 29% in the APAP group (Figure 9). Silymarin (50 µg/mL) significantly improved HepG2 cell viability in 38.2% in the APAP group.

### 2.4. Quantification In Vitro of ROS

APAP increased ROS level compared to the control in HepG2 cells. All tested extracts decreased ROS levels compared to the APAP group, showing similar ROS levels to the control and the silymarin group (Figure 10).

In post-treatment, all extracts also were able to decrease oxidative stress. However, none of them were able to improve HepG2 cell viability (Figure 11).

All extracts also decreased oxidative stress in co-treatment, showing similar ROS levels to the control (Figure 12).

## 3. Discussion

This study aimed to investigate the cytotoxicity of aqueous extracts to find safe doses and to assess their potential against APAP-induced toxicity with an assay. Like the extracts, APAP was also evaluated and demonstrated through the loss of cell viability and increase of ROS that leads to cell death due to the intoxication of liver cells.

The critical mechanism in APAP-induced hepatotoxicity is mitochondrial dysfunction. Disruptions in mitochondrial function cause oxidative stress by increasing the production of ROS, especially superoxide (O_2_^−^). This high production results in oxidative damage with loss of cell function, leading to apoptosis or necrosis [23,25].

According to Behrend et al. (2019), the APAP cytotoxicity effect in the HepG2 cell line occurs through growth inhibition via caspase-mediated apoptosis and not necrosis, as observed in primary hepatocyte cell line [26]. Choi et al. (2017) also showed the APAP toxicity mechanism in HepG2 and HepaRG cell lines. APAP toxicity was mediated by apoptosis in both cell lines [27].

Behrend et al. (2019) showed that NADPH levels are negatively affected after exposition to APAP cytotoxic doses. NADPH acts as a hydrogen donor coenzyme in reactions to protect against oxidizing compounds. The oxidative stress following APAP exposure occurs due to reduced NADPH production and not due to reactive species’ consumption [26]. The same authors also showed that APAP toxicity is independent of oxidative stress in HepG2 cells. APAP toxicity is linked to a decoupling of glycolysis from the TCA cycle, lactic acidosis, reduced NADPH production, and subsequent suppression of the anabolic pathways required for rapid growth [26].

Chowdhury et al. (2019) described that glutathione is depleted by APAP-reactive intermediate N-acetyl-p-benzoquinone-imine (NAPQI). The glutathione depletion leads to elevated phosphorylated-c-Jun N-terminal kinase (p-JNK), which further activates reactive oxygen species (ROS), initiates an inflammatory response, and finally leads to severe hepatic injury [28]. In another study using HepG2 cells, APAP (10 mmol/L) increased ROS production and negatively affected Nrf2 expression and NADPH protein levels [29].

The cytotoxicity evaluation is used as a predictive test to determine the safe dose for its use [30]. In the MTT assay, all the evaluated extracts showed dose-dependent toxicity, decreasing the number of viable cells with increasing concentration. GUAE presented the lowest toxicity among the tested extracts, being, therefore, safer since it needs higher doses to cause toxicity. GUAE, in the present study, showed significant cytotoxicity at 500 µg/mL concentration. On the other hand, HSAE presented the highest toxicity, thus requiring greater attention in its use.

The tested extracts showed lower toxicity than some Brazilian fruit extracts, as described by Malta et al. (2013) [31]. The EC_50_ values for gabiroba (*Campomanesia cambessedeana* Berg), guapeva (*Pouteria gardneriana* Radlk), and murici (*Byrsonoma verbascifolia* Rich) were 40.7 ± 4.8, 37.9 ± 2.2 mg/mL, and 173.6 ± 18.2 mg/mL in HepG2 cells [31].

In the APAP-induced toxicity assay, all extracts showed protective activity against the action of APAP in the pre-treatment, and the CCAE being the most active and showing similar protection to the positive control silymarin. CCAE protection can be explained by the presence of mangiferin identified in the chemical evaluation. Mangiferin is commonly found in the species *Mangifera indica* L., and several pharmacological activities have been described for this compound, such as antioxidant, antidiabetic, antitumor, anti-inflammatory, antibacterial, neuroprotective, cardioprotective, and analgesic effects [32,33]. Mangiferin is already known for hepatoprotective activities related to its antioxidant activity. In the study by Das et al. (2012), mangiferin treatment reduced galactosamine-induced apoptosis and necrosis [34]. Chowdhury et al. (2019) described that mangiferin improves APAP-induced liver injury by increasing GSH production and reducing APAP-CYS formation. In addition, mangiferin inhibits sustained JNK activation after APAP overdose and ameliorates oxidative stress and inflammation by the JNK pathway [28].

In the HEAE, the protection can be explained by rutin and chlorogenic acid. In the other two extracts, GUAE and HSAE, the protection activity against APAP-induced toxicity can be explained by the same compound found in HEAE, rutin. Pre-treatment with rutin has already been shown to reduce plasma transaminase activity, improve histological signs of liver damage, and exert antioxidant activity against CCl_4_ intoxication in mice [35]. Reddy et al. (2017) described the protective effect of rutin against induced hepatotoxicity in rats compared to silymarin. Rutin was superior to silymarin in restoring the pathological alterations in APAP-induced hepatotoxicity in Wistar albino rats [36]. This same hepatoprotective activity also occurs for chlorogenic acid [37,38].

In the post-treatment with the aqueous extracts, the extracts failed to reverse the APAP damage. Only silymarin showed mild activity, as well as the CCAE, at the concentration of 50 µg/mL. In Manov et al. (2004), APAP caused oxidative and apoptotic damage in HepG2 cells, while N-acetylcysteine was able to prevent oxidative damage but did not affect apoptotic damage [39]. Hao et al. (2012) used HepG2 cells and acetaminophen as toxicity-inducing agent. These authors found similar results evaluating sesquiterpenoid from *Acorus calamus* rhizome [40]. Corroborating these results, Choi et al. (2017) also showed no effect in the post-treatment with *Angelica keiskei* extract in HepG2 cells after induction of APAP toxicity. However, this extract showed a protective effect with pre-treatment [27]. Choi et al. (2017) also demonstrated that *A. keiskei* extract down-regulates apoptosis via intrinsic and extrinsic pathways against APAP-induced hepatotoxicity [27].

Considering that APAP can increase the incidence of hepatotoxicity and nephrotoxicity when used concomitantly with potentially hepatotoxic plants, this effect was not observed in this study [41]. Britza et al. (2022) observed that *Psoralea corylifolia* increased APAP toxicity [42]. Unlike that found in the study by Abdel-Ghaffar et al. (2017), in which after induction of hepatocyte toxicity with isoniazid, a tuberculosis treatment drug, co-administration with rutin could improve hepatocyte injury due to its antioxidant activities [43].

None of the aqueous extracts potentiated APAP toxicity in HepG2 cells in the co-treatment assay. Moreover, the extracts prevented APAP toxicity in 11% to 29% HepG2 cells. Again, in this treatment, CCAE showed the best response.

In the ROS quantification assay, it was observed that APAP treatment caused oxidative stress, and in all treatments, the extracts showed lower amounts of ROS compared to APAP and showed similar levels to the silymarin positive control. Possibly, the ROS levels decreased due to the presence of identified antioxidant compounds in the extracts, such as rutin, chlorogenic acid, and mangiferin.

Corroborating our study, Parikh et al. (2015), evaluating the action against APAP-induced toxicity effect on HepG2 cells, showed that extracts from *Brassica juncea* seed suppressed ROS generation when subjected to pre-treatment and post-treatment with APAP (20 mmol/L) [44]. In another study, APAP (15 mmol/L) caused severe oxidative stress with increased ROS, and isoorientin, a flavonoid, could suppress ROS production in a dose-dependent manner, involving activation of the Nrf2 antioxidant pathway [45]. Niture and Jaiswal (2012) described Nrf2 protein up-regulates antiapoptotic protein Bcl-2 and prevents cellular apoptosis [46].

Flavonoids, as secondary metabolites of plants, are considered antioxidants due to their capacity to donate hydrogen atoms to free radicals, reducing oxidative stress [47]. In the study by Alia et al. (2006), the influence of rutin and quercetin on the antioxidant defense system in HepG2 cells was evaluated. The authors observed a decreased ROS level, indicating that these flavonoids produced positive changes in the antioxidant defense system in HepG2 cells, improving oxidative stress [48]. Tabolacci et al. (2023) showed an antiapoptotic mechanism of rutin. This flavonoid reduced the high levels of ROS and increased cell viability after photo-oxidative stress in a fibroblast cell line. Rutin modulated the Nrf2 transcriptional pathway, resulting in an increase in reduced glutathione and Bcl2/Bax ratio and the subsequent protection of mitochondrial respiratory capacity [49].

In another study using HepG2 cells and APAP (10 mmol/L), quercitrin, a major constituent of *Toona sinensis* leaves, suppressed ROS production and increased Nrf2 expression and NADPH protein levels [29]. The same protective effect was demonstrated by mangiferin in mercury intoxication in HepG2 cells. Pre-treatment, besides decreasing the percentage of apoptotic cells after the mercury toxic effect, also inhibited ROS levels, restored mitochondrial membrane potential, and normalized cellular antioxidant levels [50].

These studies reinforce the present results since the extracts showed antioxidant activity in response to APAP-induced damage, possibly by the presence of phenolic compounds and flavonoids found in their composition. These extracts probably suppress ROS production, regulate the activation of the Nrf2 antioxidant pathway, regulate the antiapoptotic protein Bcl-2, and prevent cell apoptosis. However, these extracts cannot reverse the active cell apoptosis process (post-treatment).

## 4. Materials and Methods

### 4.1. Plant Material and Preparation of Extract

The access of species studied, *Cheiloclinium cognatum* (Miers) A.C.Sm. (Celastraceae), *Guazuma ulmifolia* Lam (Malvaceae), *Hancornia speciosa* Gomes (Apocynaceae), and *Hymenaea stigonocarpa* Mart. ex Hayne (Fabaceae), was registered in Sistema Nacional de Gestão do Patrimônio Genético e do Conhecimento Tradicional Associado—SisGen (Brazilian National System of Management of Genetic Heritage and Associated Traditional Knowledge) under no. A215A9A. Leaves were collected and identified by the botanist Dr. Christopher William Fagg between February and September 2018 in Brasília-DF (Brazil). Information regarding the date, time, geographic coordinates, and voucher specimens numbers are described in Table 1.

After collection, the leaves were dried and crushed. Aqueous extracts (1:10) were prepared by infusion, then filtered, frozen, and lyophilized with a vacuum of 15 mTorr and temperature of −70 °C (SP Scientific Advantage Plus XL-70, New Life Scientific, Cridersville, OH, USA).

### 4.2. Phytochemical Analyzes

#### 4.2.1. High-Performance Liquid Chromatography Analysis (HPLC)

LaChrom Elite chromatograph (Hitachi, Tokyo, Japan) equipped with L2130 pump, L2200 injector, L2455 DAD detector, and L2300 column oven were used. The column was the LiChroCART^®^ reverse-phase C18e (250 × 4.6 mm, 5 µm, Merck, Germany) coupled with a pre-column of the same characteristics. The analysis was conducted for 55 min at 25 °C with a flow rate of 0.6 mL/min. The mobile phase constitutes an HPLC-grade solvent gradient: phosphoric acid 1% (A) (Sigma-Aldrich^®^, Darmstadt, Germany) and acetonitrile (B) (Tedia^®^, Fairfield, OH, USA). The mobile phase gradient changed from 90% A and 10% B to 70% A and 30% B in 40 min. In the next 10 min, there was a change in the mobile phase of 50% A and 50% B, followed by 5 min of re-equalization to original conditions [51]. The samples were prepared at 3 mg/mL in HPLC-grade methanol and filtered through a 0.45 µm porosity (Millipore Millex^®^, Cork, Ireland) disposable filter. The chromatograms were compared with the standard library for the identification of compounds present in aqueous extracts. The chromatograms were extracted at a wavelength of 354 nm. The standards chlorogenic acid, gallic acid, catechin, epicatechin, hyperoside, myricetin, vitexin, isochlorogenic acid, neochlorogenic acid, and rutin were purchased from Sigma-Aldrich^®^ (Darmstadt, Germany). Caffeic acid, isoquercitrin, and quercetin were purchased from Cromadex^®^ (Los Angeles, CA, USA). 

#### 4.2.2. Ultra-High Performance Liquid Chromatography with Tandem Mass Spectrometry Analysis (UHPLC-MS/MS)

Samples were analyzed using a UPLC (Waters Acquity H-Class^®^, Milford, CT, USA) coupled to a photodiode arrangement detector (PDA) (Waters Acquity^®^, Milford, USA) in series with a triple quadruple mass spectrometer (Xevo^®^ TQ-XS, Waters Acquity H-Class^®^, Milford, CT, USA). The stationary phase was a UPLC BEH C18 column (Waters Acquity^®^, Milford, CT, USA) with 2.1 × 100 mm × 1.7 microns particles. The mobile phase comprised 0.1% formic acid (pump A) and acetonitrile (pump B). The column was maintained at 35 °C, and the flow of the mobile phase was 0.35 mL/min with 100% A and 0% B with the linear gradient up to 50% A and 50% B at 20 min, returning to original conditions after 4 min of re-equalization. PDA monitoring was continuous in the range of 230 to 500 nm. The mass spectrometer was operated in two different modes on separate injections. Initially, complete negative ion scans were acquired in the *m*/*z* range of 100 to 1000 every 0.4 s using cone tension. After this, directed MS/MS scans were performed using 30 V cone voltage and 40 V collision energy of the main relevant [M-H] ions. The ion source temperature was 130 °C, the desolvation gas was nitrogen at 950 L/h, and the desolvation temperature was 450 °C. In every case, the capillary voltage was 2.7 KV. The data were obtained in negative mode as it generated more data than in positive mode.

### 4.3. In Vitro APAP-Induced Toxicity Assay

#### 4.3.1. Cell Culture

Human hepatocarcinoma cells (HepG2) (ATCC: HB8065) were obtained from the Banco de Células do Rio de Janeiro (BCRJ). The cells were grown in sterile culture flasks of 75 cm^2^ containing Dulbecco’s Modified Eagle Medium (DMEM) enriched with L-glutamine (Gibco^®^, Billings, MT, USA), pyridoxine hydrochloride (Sigma-Aldrich^®^, Darmstadt, Germany), sodium pyruvate (Sigma-Aldrich^®^, Darmstadt, Germany), antibiotic solution with 1% streptomycin (Sigma-Aldrich^®^, Darmstadt, Germany), 0.6% penicillin G (10 mL/L) (Sigma-Aldrich^®^, Darmstadt, Germany), and 10% serum fetal bovine (SFB) (Gibco^®^, Billings, MT, USA). The cells were incubated (carbon dioxide incubator, Panasonic^®^, Kadoma, Japan) at 37 °C and 5% carbon dioxide (CO_2_) until 80–90% confluence. Cell subculturing was carried out with a minimum of 3 passages and a maximum of 10 passages.

#### 4.3.2. Cytotoxicity Test

Extracts’ cell cytotoxicity in HepG2 cells was evaluated with the MTT method according to Mosmann (1983) and Hansen et al. (1989) [52,53]. This test was conducted to determine the best concentrations of extracts, APAP (acetaminophen or N-(4-hydroxyphenyl)acetamide), and silymarin to be used in the APAP-induced toxicity test. The chosen APAP concentration corresponded to the IC_50_ of the cytotoxicity curve. Extracts and silymarin concentrations used in the APAP-induced toxicity test did not show cytotoxicity to HepG2 cells, except for HSAE. For this extract, the used concentration was that presenting the lowest toxicity.

APAP was evaluated in a concentration range of 1–50 nM. Extracts were evaluated in the following concentrations: *Cheiloclinium cognatum* (Miers) A.C. Sm. aqueous extract (CCAE) and *Guazuma ulmifolia* Lam. aqueous extract (GUAE), range of 50–800 µg/mL; *Hancornia speciosa* Gomes aqueous extract (HEAE), range of 25–600 µg/mL; and *Hymenaea stigonocarpa* Mart. ex Hayne aqueous extract (HSAE), range of 25–300 µg/mL. Silymarin was evaluated at a range of 25–500 µg/mL.

The analysis was performed in a 96-well plate containing 25 × 10^3^ cells/well in DMEM. After that, the plate was maintained in the CO_2_ incubator (37 °C and 5% CO_2_) for 24 h. After 24 h, DMEM was removed and replaced by the treatment (extract or APAP or silymarin all solubilized in DMEM). After 24 h, the medium was removed, and 50 µL of MTT (Sigma^®^) diluted in DMEM without phenol red (Gibco^®^, Billings, MT, USA) (1 mg/mL) was added. Then, the plate was incubated at 37 °C and 5% CO_2_ for 4 h. At the end of this period, 150 µL of acidified isopropanol was added to the wells. The plate was forwarded to the ELISA plate reader (PerkinElmer^®^, Waltham, MA, USA), and the absorbance was performed at 570 nm. The absorbances obtained for each well were used to calculate the cell viability percentage and the IC_50_ values. The cell viability percentage was calculated by comparing the absorbance of the groups that received the treatments and the absorbance of the control group. The results were expressed by mean of the triplicates in three independent assays.

#### 4.3.3. Activity against APAP-Induced Toxicity

This test was performed according to MTT method described by Mosmann (1983) [52] and Hansen et al. (1989) [53]. The treatments used were similar to those described by Choi et al. (2017) [27]. The assay was performed in a 96-well plate with a 25 × 10^3^ cells/well density. After 24 h of plating, three treatment protocols were performed: pre-treatment, post-treatment, and co-treatment. In the pre-treatment protocol, cells were exposed to extracts or silymarin (positive control) for 24 h. After that, cells were exposed to APAP for more 24 h. In the post-treatment protocol, cells were exposed to APAP for 24 h. After that, cells were exposed to extracts or silymarin for more 24 h. In the co-treatment protocol, cells were exposed to APAP and extracts or APAP and silymarin for 48 h. At the end of the 48 h of treatment in the three protocols, the medium was removed, and cell viability was determined following the same procedure previously described in Section 4.3.2 about the cytotoxicity test.

#### 4.3.4. In Vitro Quantification of Reactive Oxygen Species (ROS)

The same treatment protocols described in Section 4.3.3. were carried out. At the end of 48 h, cells were washed with 50 µL of PBS. Then, 100 µL of 2,7-dichlorofluorescein diacetate (DCFH-DA) (20 µM) was added. The plate was incubated in the dark at 37 °C for 30 min. After this time, DCFH-DA was removed, and 50 µL of PBS was added. Fluorescence reading was performed at 485 nm for excitation and 535 nm for emission using an ELISA plate reader (PerkinElmer^®^). Intracellular ROS levels were normalized by quantifying proteins and were determined using the Pierce Kit™ BCA Protein Assay (ThermoFisher Scientific^®^, Waltham, MA, USA).

### 4.4. Statistical Analysis

The software used for the statistical analyses was GraphPad Prism^®^ Version 6.0. The data distribution was evaluated. For the normal distribution, data were applied to parametric ANOVA, Dunnet test, and data were represented by mean and standard deviation. Non-normal distribution data were analyzed with Kruskal–Wallis-Dunn’s nonparametric test, and data were represented by median and interquartile range. Significant differences were considered when *p* values < 0.05.

## 5. Conclusions

The study demonstrated the presence of phenolic compounds in the extracts. Rutin was identified in HEAE, HSAE, and GUAE. In HEAE, besides rutin, chlorogenic acid was identified, which had already been reported for the species. In the CCAE, mangiferin was identified. All extracts showed dose-dependent toxicity; CCAE was the least toxic aqueous extract, and HSAE was the most toxic. In the activity against APAP-induced toxicity, the extracts protected HepG2 cells against APAP toxicity and may be candidates for supplements that could be used to prevent liver damage. In the concomitant treatment of the extracts with APAP, it was demonstrated that the extracts do not present a synergistic toxic effect, with no potentiation of occurring toxicity. Regarding the treatment in which the extracts are applied after the toxic action of APAP, it was demonstrated that they cannot reverse the damage, reinforcing the apoptotic mechanism of APAP in HepG2 cells with substantial growth inhibition. The HepG2 model was helpful in screening plant species that can also be tested against other toxic compounds and in other experimental models since the extracts showed considerable cytoprotective effects and important antioxidant characteristics.

## Figures and Tables

**Figure 1 plants-12-03393-f001:**
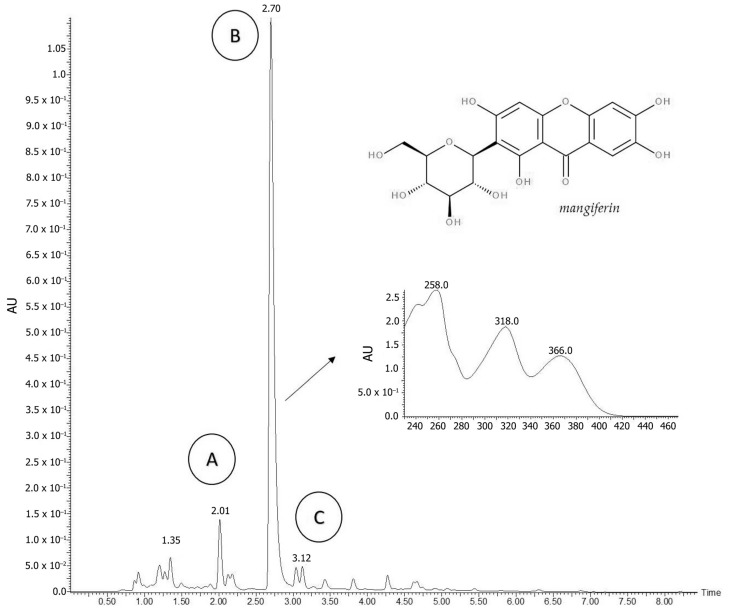
Chromatographic profile of the *C. cognatum* aqueous extract (CCAE) with UPLC/MS/MS technique at wavelength of 354 nm. A: Peak 1 (tR 2.01 min; UV Max 230.0, 295.0). B: Peak 2 (tR 2.70 min; UV Max 258.0, 318.0, 366,0), strongly suggestive of a mangiferin compound. C: Peak 3 (tR 3.12 min; UV Max 230.0, 274.0, 337.0), suggestive of a catechin or epicatechin compound.

**Figure 2 plants-12-03393-f002:**
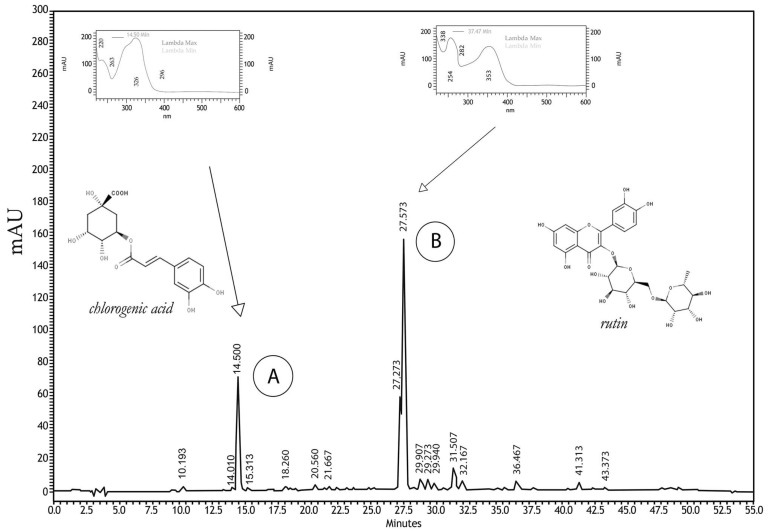
Chromatographic profile of the *H. speciosa* aqueous extract (HEAE) with HPLC/DAD technique at the wavelength of 354 nm. A: peak with retention of 14.5 min identified as chlorogenic acid and its respective UV spectrum (λmax: 326,233; purity: 1.00). B: peak with retention time of 27.6 with similarity to rutin with its respective UV spectrum (λmax: 256,353; purity: 1.00).

**Figure 3 plants-12-03393-f003:**
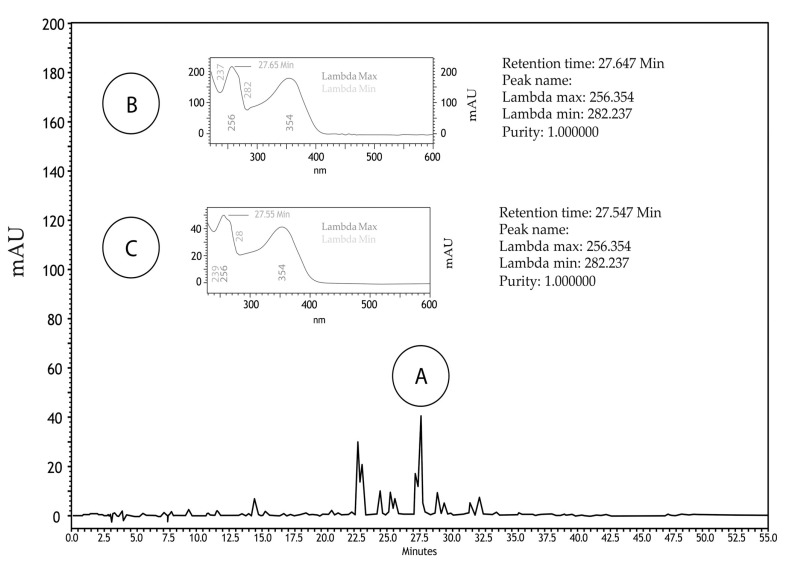
Chromatographic profile of the *G. ulmifolia* aqueous extract (GUAE) with HPLC/DAD technique at wavelength of 354 nm. A: peak detected in the GUAE showing a retention time of 27.547 min (similarity index: 0.9973; purity: 1.00; λmax: 256,354). B: spectrum of the rutin standard used for comparison. C: spectrum of the peak detected in the GUAE.

**Figure 4 plants-12-03393-f004:**
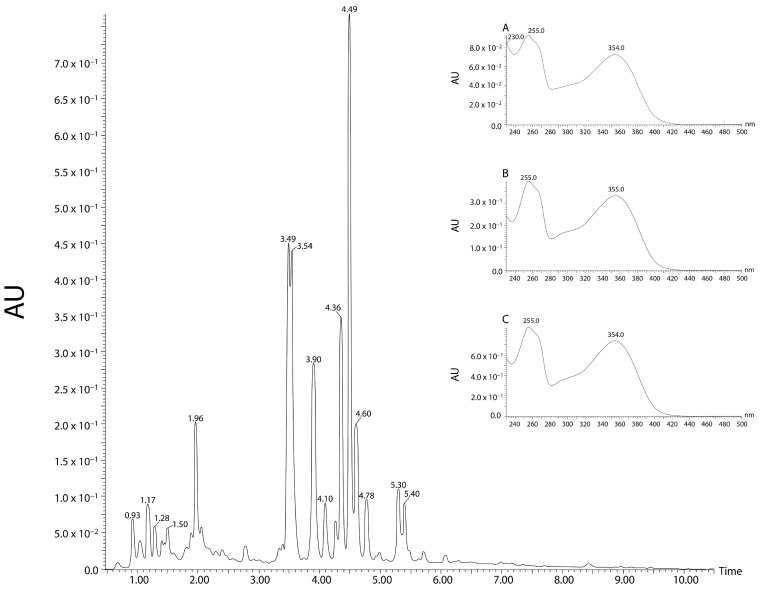
Chromatographic profile of the *G. ulmifolia* aqueous extract (GUAE) with UHPLC/DAD/MS/MS technique at wavelength of 354 nm. The peaks identified showed similarity to rutin. A: spectrum of the peak with a retention time of 4.10 min; B: spectrum of the peak with a retention time of 4.36 min; and C: spectrum of the peak with a retention time of 4.49 min.

**Figure 5 plants-12-03393-f005:**
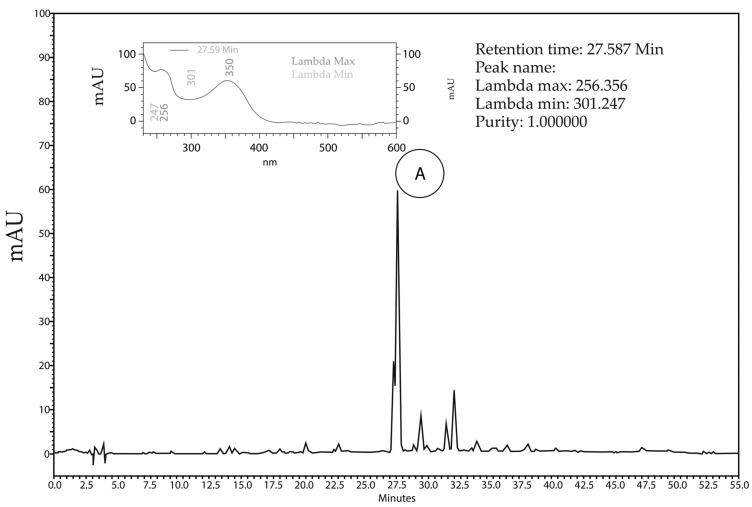
Chromatographic profile of the *H. stigonocarpa* aqueous extract (HSAE) with HPLC-DAD technique at wavelength of 354 nm. A: Peak with retention time of 27.6 min and its respective UV spectrum (purity: 1.00; λmax: 255,354).

**Figure 6 plants-12-03393-f006:**
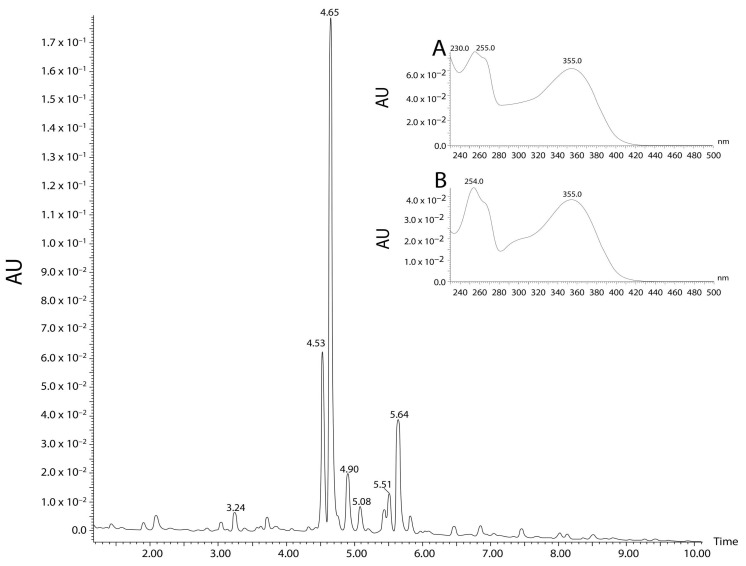
Chromatographic profile of the *H. stigonocarpa* aqueous extract (HSAE) with UHPLC/DAD/MS/MS at wavelength of 354 nm. The peaks in 4.53 min and 4.65 min showed similarity to rutin. A: spectrum of peak with a retention time of 4.53 min; and B: spectrum of the peak with a retention time of 4.65 min.

**Figure 7 plants-12-03393-f007:**
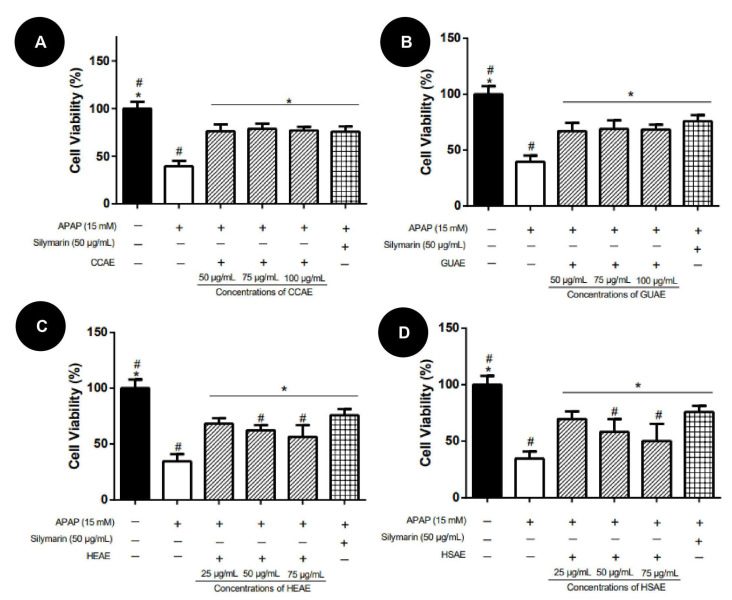
HepG2 cell viability after 24 h of pre-treatment with aqueous extracts and toxicity induction with APAP (15 mM). Cell viability values (%) are expressed as mean and standard deviation (*n* = 3). The + and − signs correspond to the presence and absence of treatment, respectively. The first column corresponds to the control without any treatment. The second column corresponds to the group exposed to APAP only. The third, fourth and fifth column correspond to the treatment with different concentration extract. The sixth column corresponds to the silymarin treatment after exposed to APAP (positive control). (**A**): Pre-treatment results for aqueous extract of *C. cognatum* (CCAE). (**B**): Pre-treatment results for aqueous extract of *G. ulmifolia* (GUAE). (**C**): Pre-treatment results for aqueous extract of *H. speciosa* (HEAE). (**D**): Pre-treatment results for aqueous extract of *H. stigonocarpa* (HSAE). Positive hepatoprotection control corresponds to silymarin at a dose of 50 µg/mL (sixth column in the graph). The results of the viability of the different treatments were compared with the APAP and with silymarin using analysis of variance (ANOVA) with Dunnet’s post-test (*, *p* ≤ 0.05, significantly different from APAP) (#, *p* ≤ 0.05, significantly different from silymarin).

**Figure 8 plants-12-03393-f008:**
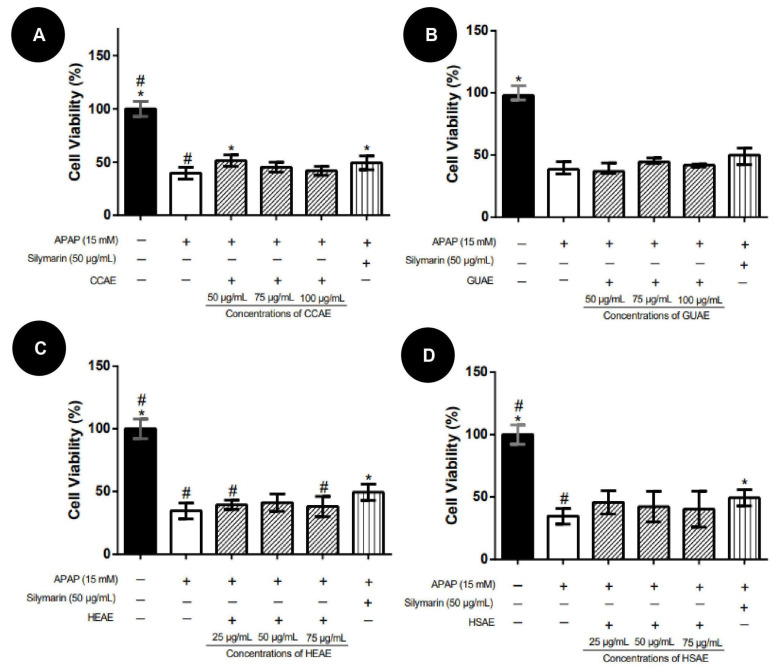
HepG2 cell viability after 24 h of induction APAP (15 mM) toxicity and post-treatment with aqueous extracts. Cell viability values (%) are expressed as median and inter-quartile range (*n* = 3). The + and − signs correspond to the presence and absence of treatment, respectively. The first column corresponds to the control without any treatment. The second column corresponds to the group exposed to APAP only. The third, fourth and fifth column correspond to the treatment with different concentration extract. The sixth column corresponds to the silymarin treatment after exposed to APAP (positive control). (**A**): Post-treatment results for aqueous extract of *C. cognatum* (CCAE). (**B**): Post-treatment results for aqueous extract of *G. ulmifolia* (GUAE). (**C**): Post-treatment results for aqueous extract of *H. speciosa* (HEAE). (**D**): Post-treatment results for aqueous extract of *H. stigonocarpa* (HSAE). Positive hepatoprotection control corresponds to silymarin at a dose of 50 µg/mL (sixth column in the graph). The results of the viability of the different treatments were compared with APAP and silymarin using Kruskal–Wallis with Dunn’s post-test (* *p* ≤ 0.05 significantly different from APAP) (#, *p* ≤ 0.05, significantly different from silymarin).

**Figure 9 plants-12-03393-f009:**
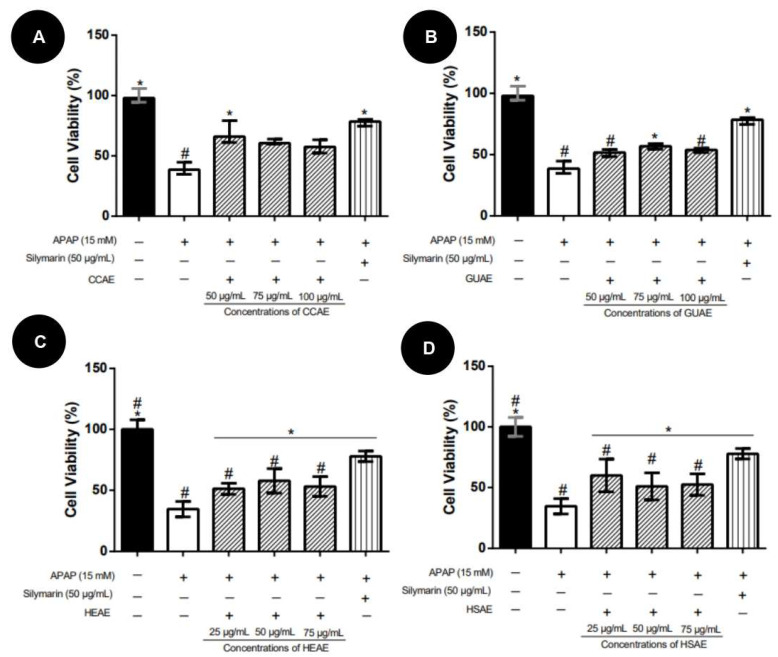
HepG2 cell viability after 48 h of co-treatment with aqueous extracts and APAP (15 mM). Cell viability values (%) are expressed as median and inter-quartile range (*n* = 3). The + and − signs correspond to the presence and absence of treatment, respectively. The first column corresponds to the control without any treatment. The second column corresponds to the group exposed to APAP only. The third, fourth and fifth column correspond to the treatment with different concentration extract. The sixth column corresponds to the silymarin treatment after exposed to APAP (positive control). (**A**): Co-treatment results for aqueous extract of *C. cognatum* (CCAE). (**B**): Co-treatment results for aqueous extract of *G. ulmifolia* (GUAE). (**C**): Co-treatment results for aqueous extract of *H. speciosa* (HEAE). (**D**): Co-treatment results for aqueous extract of *H. stigonocarpa* (HSAE). Positive hepatoprotection control corresponds to silymarin at a dose of 50 µg/mL (sixth column in the graph). The results of the viability of the different treatments were compared with APAP and silymarin using Kruskal–Wallis with Dunn’s post-test (* *p* ≤ 0.05 significantly different from APAP) (#, *p* ≤ 0.05, significantly different from silymarin).

**Figure 10 plants-12-03393-f010:**
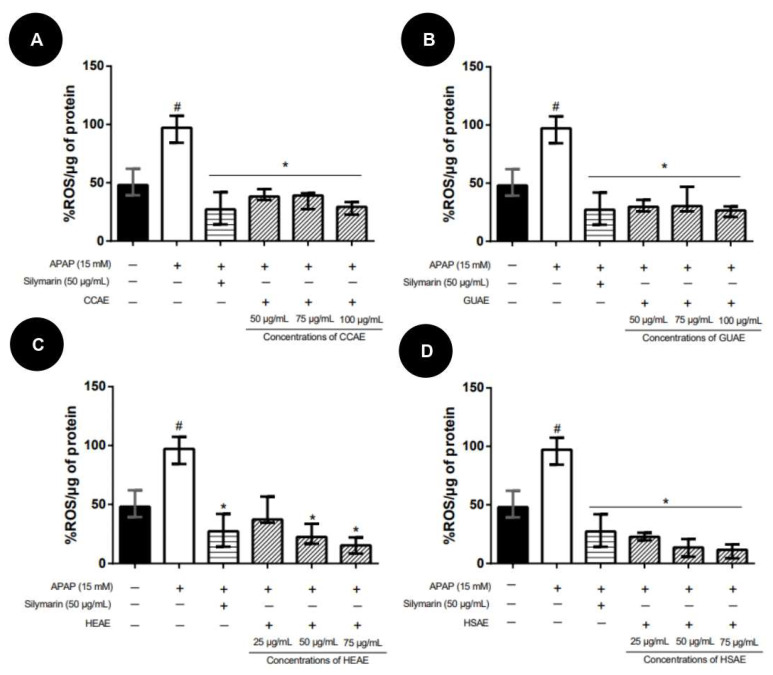
ROS levels after pre-treatment protocol with the aqueous extracts and APAP (15 mM). The results are expressed as median and inter-quartile range (*n* = 3). The + and − signs correspond to the presence and absence of treatment, respectively. The first column corresponds to the control without any treatment. The second column corresponds to the group exposed to APAP only. The third, fourth and fifth column correspond to the treatment with different concentration extract. The sixth column corresponds to the silymarin treatment after exposed to APAP (positive control). (**A**): Pre-treatment results for aqueous extract of *C. cognatum* (CCAE). (**B**): Pre-treatment results for aqueous extract of *G. ulmifolia* (GUAE). (**C**): Pre-treatment results for aqueous extract of *H. speciosa* (HEAE). (**D**): Pre-treatment results for aqueous extract of *H. stigonocarpa* (HSAE). Positive control corresponds to silymarin at a dose of 50 µg/mL (third column in the graph). The results of the different treatments were compared with APAP and silymarin using Kruskal–Wallis with Dunn’s post-test (* *p* ≤ 0.05 significantly different from APAP) (#, *p* ≤ 0.05, significantly different from silymarin).

**Figure 11 plants-12-03393-f011:**
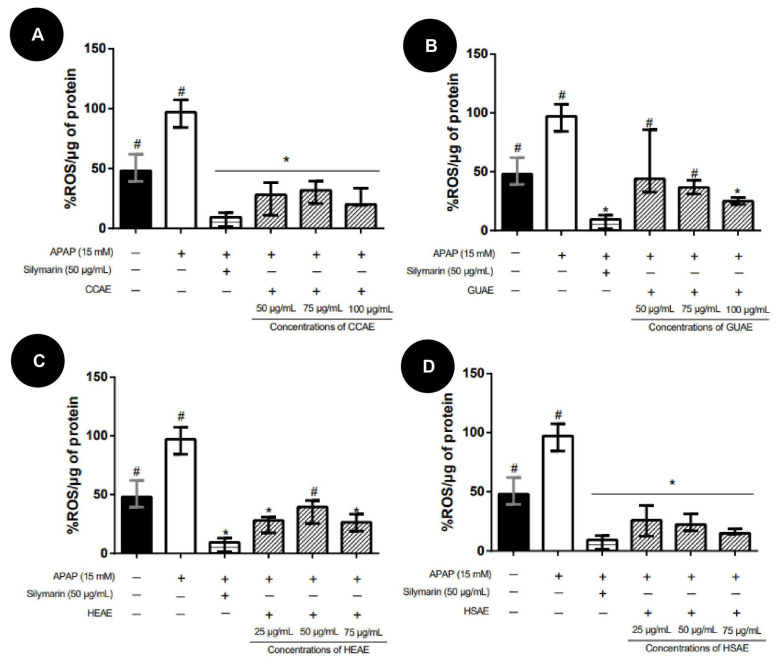
ROS levels after post-treatment protocol with the aqueous extracts and APAP (15 mM). The results are expressed as median and inter-quartile range (*n* = 3). The + and − signs correspond to the presence and absence of treatment, respectively. The first column corresponds to the control without any treatment. The second column corresponds to the group exposed to APAP only. The third, fourth and fifth column correspond to the treatment with different concentration extract. The sixth column corresponds to the silymarin treatment after exposed to APAP (positive control). (**A**): Post-treatment results for aqueous extract of *C. cognatum* (CCAE). (**B**): Post-treatment results for aqueous extract of *G. ulmifolia* (GUAE). (**C**): Post-treatment results for aqueous extract of *H. speciosa* (HEAE). (**D**): Post-treatment results for aqueous extract of *H. stigonocarpa* (HSAE). Positive control corresponds to silymarin at a dose of 50 µg/mL (third column in the graph). The results of the different treatments were compared with APAP and silymarin using Kruskal–Wallis with Dunn’s post-test (* *p* ≤ 0.05 significantly different from APAP) (#, *p* ≤ 0.05, significantly different from silymarin).

**Figure 12 plants-12-03393-f012:**
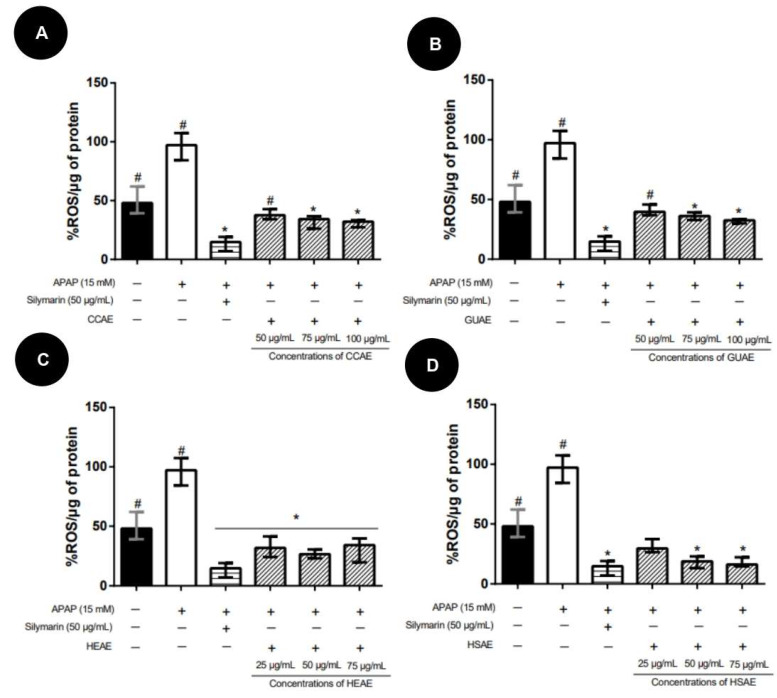
ROS levels after co-treatment protocol with the aqueous extracts and APAP (15 mM). The results are expressed as median and inter-quartile range (*n* = 3). The + and − signs correspond to the presence and absence of treatment, respectively. The first column corresponds to the control without any treatment. The second column corresponds to the group exposed to APAP only. The third, fourth and fifth column correspond to the treatment with different concentration extract. The sixth column corresponds to the silymarin treatment after exposed to APAP (positive control). (**A**): Co-treatment results for aqueous extract of *C. cognatum* (CCAE). (**B**): Co-treatment results for aqueous extract of *G. ulmifolia* (GUAE). (**C**): Co-treatment results for aqueous extract of *H. speciosa* (HEAE). (**D**): Co-treatment results for aqueous extract of *H. stigonocarpa* (HSAE). Positive control corresponds to silymarin at a dose of 50 µg/mL (third column in the graph). The results of the different treatments were compared with APAP and silymarin using Kruskal–Wallis with Dunn’s post-test (* *p* ≤ 0.05 significantly different from APAP) (#, *p* ≤ 0.05, significantly different from silymarin).

**Table 1 plants-12-03393-t001:** General information about the collection of plant species.

Species	Date	Time	Place	Geographic Coordinates	Exsiccata Numbers
*Cheiloclinium cognatum*	9 March 2018	10 a.m.–12 p.m.	(1)	15°90′85.83″ S 47°91′36.11″ W	Mendonça, R. 4991
*Guazuma ulmifolia*	2 May 2018	17 p.m.–18 p.m.	(1)	15°52′30.90″ S 47°57′24″ W	Fagg, C.W. 2484
*Hancornia speciosa*	22 February 2018	10 a.m.–11 a.m.	(2)	15°45′48.21″ S 47°51′50.67″ W	Fagg, C.W. 2495
*Hymenaea stigonocarpa*	17 May 2018	14 p.m.–15 p.m.	(2)	15°45′48.21″ S 47°51′50.67″ W	Fagg, C.W. 2491

(1) Água Limpa Farm, Park Way, Brasilia, Brazil. (2) University of Brasilia, Darcy Ribeiro campus, Brasilia, Brazil.

## Data Availability

Not applicable.

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
