# Peer review of "Hepatoprotective Effects of Four Brazilian Savanna Species on Acetaminophen-Induced Hepatotoxicity in HepG2 Cells"

_plants, 2023, doi:10.3390/plants12193393_

Round 1
Reviewer 1 Report
1. References are required in line 66-68.
2. Typo errors in line 78. “Hancornia speciosa Gomes e Hymenaea stigonocarpa Mart. Ex Hayne” It looks like one plant.
3. Did you analyze other peaks in GUAE? According to Fig 3 &4, only one peak was suggested as rutin. Is there any plan to analyze other major peaks of GUAE and HSAE?
4. In supplementary Fig 3, a typo error is found in X axis. Please change from 0,05 to 0.05.
5. How were the experiments conducted? Please explain the difference of the methods between pre, post and co-treatment in detail in Materials and methods.
6. Please explain how you decide the final concentration of each plant in MTT assay? If you worried about IC50, the concentrations of each plant are totally different. But you choose same concentration with CCAE and GUAE, and HEAE and HSAE, respectively.
7. In line 272-273, a typo error is found. I think this is one sentence, isn’t it?
8. Please check the format of figures. I think A, B, C, and D in Figures not 1, 2, 3, and 4.
9. How did each plant protect liver cells from toxicity of APAP? Please explain possible mechanisms in each plant and provide the proofs to support the mechanisms. You mentioned the action of APAP in the discussion. What kinds of APAP action was suppressed by those 4 plants? For example, Nrf2 was increased by CCAE so that ROS was decreased.
10. In spite of reduction of ROS, why was cell viability not recovered by 4 plants in post and co-treatment?
Please check typo errors.
Reviewer 2 Report
In the manuscript prepared by Y. M. Fonseca-Bazzo, hepatoprotective effects of four Brazilian savanna species on acetaminophen-induced hepatotoxicity in HepG2 cells have been introduced. The presence of phenolic compounds and flavonoids such as chlorogenic acid, hyperoside, isoquercitrin, rutin and mangiferin were confirmed by UPLC/MS/MS technique. Although the similarity over 99% were mentioned, the 1H NMR and 13C NMR spectra of each compound are essential in my opinion. I am not sure whether the hepatoprotective effects of each flavonoid compound mentioned above have been investigated or not. If no, please make a comparison.
the extracts contain known compounds which were confirmed only by UPLC/MS/MS technique,please add the 1H NMR and 13CNMR analyses.
Reviewer 3 Report
Dear authors,
Comments
1. I feel that the use of the abbreviation (APAP) is not reflecting the drug name "Acetaminophen" or even the paracetamol. You may better not abbreviate the drug name or use other more connected abbreviations.
2. I also feel that the following introductory sentences should be deleted from the abstract: "Acetaminophen (APAP) is a widely used drug that, in therapeutic doses, is considered safe but, in high doses, presents a hepatotoxicity risk. In the USA, APAP overdose cases correlate with acute hepatic insufficiency with a consequent need for transplantation. Medicinal plants may present compounds that are a source for the discovery of drugs that act as either monotherapy or coadjuvants to limit acetaminophen toxicity."
3. Also, the following sentence, "Hepatoprotection was evaluated using pre-treatment, co-treatment, and post-treatment protocols." should be completed as "Hepatoprotection was evaluated using different protocols, i.e., pre-treatment, co-treatment, and post-treatment of the cells with the acetaminophin and the plants extracts."
4. You may need to add refs after the following sentence: "generation to generation and also due to public interest in natural therapies.". I recomend the following updated and related citations: https://doi.org/10.1016/j.arabjc.2022.103950; https://doi.org/10.3390/ijms23042149
5. Authors should explain why they used concentrations of 50, 75, and 100 micrograms/mL in the hepatoprotective protocols. Why they did not use smaller doses like 10 or 25 micrograms per mL.
6. Based on your experiments, please also discuss the effects of the doses of the extracts used on the cells viability in the absence of the hepatocellular toxic agent.
7. I also recommend naming the post-treatment protocol a treatment strategy and not a protective strategy against paratcetamol-induced hepatotoxicity.
8. Figures need improvement.
Regards
Minor English revision is required.
Round 2
Reviewer 1 Report
1. Enough discussion was added for possible mechanisms of each plant. However, re-organization of discussion is required. Too many paragraphs and repeat of same comments from lots of references make readers confusing. For example, in line 329-330, authors mentioned the inducible activity of APAP, and mentioned again in line 331-332, 382-383 and 384-385. I recommend one paragraph shows one topic. For example, one paragraph shows the toxicity and mechanism of APAP, the other paragraph for CCAE, and one is for the explanation why post-treatment of extracts has no effect on viability.
Please check typo errors.
Reviewer 2 Report
I have no further questions on the manuscript. It looks good.
Author Response
We appreciated the referee's opinion concerning the paper.
Reviewer 3 Report
Thanks
Author Response

(The authors gave the same response as above.)
